# Co-Amorphous Versus Deep Eutectic Solvents Formulations for Transdermal Administration

**DOI:** 10.3390/pharmaceutics15061710

**Published:** 2023-06-12

**Authors:** Yannick Guinet, Laurent Paccou, Alain Hédoux

**Affiliations:** UMR 8207–UMET–Unité Matériaux et Transformations, Université de Lille, CNRS, INRAE, Centrale Lille, F-59000 Lille, France; yannick.guinet@univ-lille.fr (Y.G.); laurent.paccou@univ-lille.fr (L.P.)

**Keywords:** low-frequency Raman spectroscopy, co-amorphous, hydrogen-bonding, deep-eutectic solvents, cryo-milling

## Abstract

Transdermal administration can be considered as an interesting route to overcome the side-effects inherent to oral intake. Designing topical formulations with maximum drug efficiency requires the optimization of the permeation and the stability of the drug. The present study focuses on the physical stability of amorphous drugs within the formulation. Ibuprofen is commonly used in topical formulations and then was selected as a model drug. Additionally, its low Tg allows easy, unexpected recrystallization at room temperature with negative consequence on skin penetration. In this study, the physical stability of amorphous ibuprofen was investigated in two types of formulations: (i) in terpenes-based deep eutectic solvents (DES) and (ii) in arginine-based co-amorphous blends. The phase diagram of ibuprofen:L-menthol was mainly analyzed by low-frequency Raman spectroscopy, leading to the evidence of ibuprofen recrystallization in a wide range of ibuprofen concentration. By contrast, it was shown that amorphous ibuprofen is stabilized when dissolved in thymol:menthol DES. Forming co-amorphous arginine–ibuprofen blends by melting is another route for stabilizing amorphous ibuprofen, while recrystallization was detected in the same co-amorphous mixtures obtained by cryo-milling. The mechanism of stabilization is discussed from determining T_g_ and analyzing H-bonding interactions by Raman investigations in the C=O and O–H stretching regions. It was found that recrystallization of ibuprofen was inhibited by the inability to form dimers inherent to the preferential formation of heteromolecular H-bonding, regardless of the glass transition temperatures of the various mixtures. This result should be important for predicting ibuprofen stability within other types of topical formulations.

## 1. Introduction

During the early stages of the drug development process, scientists are often faced with a set of challenges to provide the target product profile. In this context, bioavailability can be considered as a major concern. Many new drug candidates synthesized in the crystalline state are poorly soluble in water, with inherent low bioavailability. Another problem comes from the possible polymorphic phase transformations that the crystal can undergo subjected to various constraints inherent to the drug manufacturing process [1,2,3]. It is now well recognized that converting crystal into the amorphous state significantly improves the aqueous solubility and the drug efficacy [4,5]. Unfortunately, the amorphous state is thermodynamically unstable and unexpected crystallization can occur during storage, manufacturing processes, or administration [6]. In a first step, solid dispersions were developed with the aim of increasing the glass transition temperature (T_g_) by mixing the drug with a polymer of high T_g_ [7,8]. Ensuring the stability of the amorphous drugs within the polymer carrier needs the knowledge of the solubility curves, which requires specific experimental methods [9,10] that are not easy to implement. Indeed, there is a lack of standardized method for measuring drug/polymer solubilities, mainly because of the high viscosity of polymers, which makes solubility equilibrium difficult to reach and the determination of solubility curves very time-consuming. Since the late 2000s, a new strategy was developed to produce binary amorphous mixtures composed of two small molecular weight compounds [11,12,13], ensuring drug stability. Amino acids were recognized as co-amorphous stabilizers and promising excipients in co-amorphous formulations [14], because of their capabilities to form strong solid-state interactions with co-formers, including H-bonding, π-π, or ionic interactions [15]. These formulations can be easily prepared [16], mostly by ball milling [14], but also by classical manufacturing processes such as spray drying [17]. Despite numerous investigations on co-amorphous formulations [16] aiming to analyze the preparation methods, the capabilities of amino acids (AAs) to form co-amorphous blends, there are only a few studies focusing on the molecular interactions between co-formers reported in the literature [18], and no general description of physical mechanisms responsible for the stability of the co-amorphous blends has been proposed. Intriguingly, it was found that T_g_ of indomethacin–arginine amorphous blends was higher than that of both individual components [18]. Such a behavior has been considered as resulting from an amorphous salt formation during milling. However, there is no information about the mechanism of salt formation and the physical properties of the individual component inherent to salt formation.

Another strategy, also based on binary mixtures, has been developed since the late 1990s to design eutectic systems. In contrast to co-amorphous blends, it is recognized that the formation of eutectic systems results from the association of hydrogen-bond donor (HBD) with hydrogen-bond acceptor (HBA) molecules. The main difference with co-amorphous blends is that the active pharmaceutical ingredient (API) is systematically in the liquid state at room temperature. The formation of eutectic systems was developed for transdermal delivery of ibuprofen (IBP) [14,18,19] associated with terpenes (menthol, thymol, etc.) serving as permeation enhancers. Maintaining the mixture at temperatures above the eutectic temperature ensures the stability of amorphous APIs within the liquid formulations. This stability should not be subjected to temperature fluctuations because of the extreme difficulty of recrystallization around the eutectic point in relation with the specific microstructure of the eutectic composition [19]. In this context, it is useful to prepare formulations close to eutectic compositions for maximum decrease in the melting temperature, which then requires accurate determination of the binary phase diagram. This condition has motivated the recent revision of the phase diagram based on racemic ibuprofen (IBP) with L-menthol (M) which revealed the absence of eutectic reaction in the L-menthol-rich side [20]. More specifically, it is possible to design eutectic systems with eutectic temperature well below that predicted for ideal liquid mixtures [7,9], using natural and green solvents to enhance dissolution of APIs and to improve their bioavailability [2,21]. This class of solvents, called natural Deep Eutectic Solvents (DES), can be composed of one API [1,6], or can be used to dissolve a poorly water-soluble API [1,4]. It was shown that IBP solubility in aqueous solution can be significantly increased by using green co-solvents such as polyols and others (propylene glycol, polyethylene glycol, etc.) [22]. As a consequence, IBP solubility in DES designed from these co-solvents is also increased compared to IBP solubility in water [1]. It was also shown that IBP can be dissolved in large proportion (20 wt%) in the xylitol–citric acid DES and can be maintained in the amorphous state on a wide temperature range [4].

Despite the common use of IBP in topical formulations [23,24,25,26], the physical stability of IBP was not investigated. However, IBP easily recrystallizes [27] at room temperature with a negative effect on skin penetration. The present study aims to determine the most stable formulations among the strategies presented above for placing and maintaining a significant amount of IBP in the amorphous state, i.e., either (i) by forming co-amorphous blends or (ii) by designing eutectic mixtures or dissolving IBP in deep eutectic solvents. In the first case, the eutectic mixture ibuprofen–menthol was analyzed to better describe the controversial phase diagram [14,18]. Dissolution of ibuprofen in the thymol–L-menthol mixture, previously recognized as DES [10], was also analyzed since both components of the eutectic mixture are considered as permeating enhancers. In the second case, L-arginine (ARG) was selected for preparing co-amorphous blends, given the well-known capabilities of this AA to form co-amorphous blends [17,28]. In both cases, special attention was focused on molecular interactions between IBP and co-formers in order to understand the physical properties of the different types of mixtures responsible for the stability of amorphous IBP. Combined analyzes of molecular interactions in eutectic mixtures and in co-amorphous blends were discussed for providing a better understanding the stabilization mechanism of amorphous IBP.

These investigations were performed by using a single experimental tool, namely Raman spectroscopy in a wide spectral region covering the low-frequency region, very sensitive to the physical state of molecular materials, the mid- and high-frequency regions more sensitive for the detection and the analysis of H-bonding [29].

## 2. Materials and Methods

### 2.1. Materials

Racemic ibuprofen (designated herein as IBP) was purchased from Sigma (CAS number 15687-27-1, purity ≥99.8% GC assay) and was used without further purification. L-menthol (M) was purchased from Sigma (99.5% purity) and used as received. L-arginine (ARG, C_6_H_14_N_4_O_2_, purity ≥98%) was purchased from Sigma Aldrich and used as received.

### 2.2. Methods

#### 2.2.1. Preparation of Co-Amorphous Blends by Cryogenic Ball Milling

Mixture blends (IBP:ARG) were prepared by cryo-milling for 45 min at 30 Hz using Retsch CryoMill. Mixtures of typical mass of 1 g were milled at −196 °C. IMC and ARG were placed in a ZrO_2_ jar and milled using one ball (Ø = 20 mm) of the same material. A procedure alternating milling periods of 5 min with pause periods (milling at 5 Hz) of 1 min was used to limit mechanical heating.

#### 2.2.2. Preparation of Crystallized IBP:M Mixtures

The various molar fraction mixtures were firstly heated to obtain homogeneous liquid mixtures, and directly quenched in liquid nitrogen in order to produce cracks in the undercooled liquid mixtures. The mixtures were then kept at −10 °C for 1 to 7 days depending on the composition. The physical state of mixtures was determined from low-frequency Raman spectroscopy.

#### 2.2.3. Raman Spectroscopy

Low-frequency Raman spectroscopy (LFRS) experiments were performed on the high-dispersive XY-Dilor spectrometer composed of three gratings configured with a focal length of 800 mm, and using the 660 nm line of a Cobolt laser. Opening the slits at 150 µm makes it possible to detect a Raman signal down to 5 cm^−1^ in high resolution configuration (lower than 1 cm^−1^). All samples were loaded in spherical Pyrex cells and hermetically sealed. The temperature of each sample was regulated using an Oxford nitrogen flux device that keeps temperature fluctuations within 0.1 °C. Low-frequency Raman spectra were collected between 5 and 200 cm^−1^ in 1 min, in situ, during the heating ramp at 1 °C/min. It is well known [29] that the LFRS requires specific data processing because of the high sensitivity of the low frequency to thermal fluctuations through the Bose factor. The Raman intensity ($I_{Raman}\left(\omega ,T\right)$) was firstly transformed into reduced intensity ($I_{r}\left(\omega \right)$) according to [29,30]
$$I_{r}\left(\omega \right)=\frac{I_{Raman}\left(\omega ,T\right)}{\left[n\left(\omega ,T\right)+1\right].\omega }$$
where $n\left(\omega ,T\right)$ is the Bose factor. The low-frequency spectrum represented in reduced intensity is dominated by a very intense component, named quasielastic scattering, detected in the very low-frequency range (<50 cm^−1^), reflecting local rapid motions [31], or monomolecular reorientations as observed in rotator phases [32]. This representation of the low-frequency Raman spectrum is very well suited for detecting disorder traces via the increase in the very low-frequency intensity corresponding to the quasielastic scattering intensity (I_QES_). It was used for recognizing the absence of amorphous component and for detecting the first traces of melting in IBP:M mixtures. The reduced intensity was also used for determining T_g_ values of IBP:M mixtures and IBP-ARG co-amorphous formulations from the change in the slope detected in the I_QES_(T) plots [27]. LFRS is used for giving a structural description of molecules from the short-range order (in the amorphous state) to the long-range order (crystalline state), and most importantly makes it possible to detect and identify nanocrystalline signatures within an amorphous matrix [33,34].

The second part of Raman investigations was focused on the mid- and high-frequency domain covering the C=O stretching region between 1550 and 1750 cm^−1^ and the O/C–H stretching region lying between 2900 and 3700 cm^−1^. Spectra were collected using the InVia Renishaw spectrometer. The 785 nm line emitted from a Fandango Cobolt laser was focused on the powder sample via an achromatic lens for analyzing the largest possible volume of material (about 1 mm^3^). The sample temperature was controlled by placing the sample in a THMS 600 Linkam temperature device. The acquisition time of each spectrum was 1 min, and they were collected in situ during heating ramps at 1 °C/min. Analyzing the C=O and O–H stretching regions makes it possible to detect molecular associations via C=O…H bonding.

#### 2.2.4. Differential Scanning Calorimetry (DSC)

Experiments were performed with the DSC 3 star system analysis of Mettler Toledo equipped with the immersion cooler Huber TC100. The samples (typical mass of 10 mg) were systematically placed in an aluminum pan hermetically sealed and were flushed with highly pure nitrogen gas. All data were collected upon heating ramps performed at 1 °C/min.

## 3. Results

### 3.1. Exploring the Phase Diagram of the Binary Mixture Ibuprofen–Menthol (IBP-M)

The phase diagram of the ideal liquid phase model was plotted (in Figure 1) from the consideration that the solid–liquid equilibrium lines of each component are described by the equation [35]:$$\ln\left(x_{i}\gamma_{i}\right)=\frac{∆H_{m,i}}{R}.\left(\frac{1}{T_{m,i}}-\frac{1}{T}\right)$$
where *x_i_*, *γ_i_* are the mole fraction and the activity coefficient of the component *i*, and Δ*H_m*,*i_*, *T_m*,*i_* the enthalpy and temperature of melting. In the frame of the ideal liquid phase model activity coefficients of the two components are set to unit (*γ_i_* = 1). The phase diagram plotted in Figure 1 shows that the eutectic composition corresponds to the mole fraction of IBP, X_IBP_ = 0.25.

#### 3.1.1. Experimental Determination of the Phase Diagram from LFRS and DSC Investigations

The crystallized IBP_x_:M_(1−x)_ mixtures were systematically analyzed by step of x = 0.1 upon heating ($\dot{T}=1\;{}^{\circ}C/min$) from −50 °C up to 80 °C, and directly quenched down to −100 °C for analyzing the temperature behavior (T_g_, recrystallization) of vitrified mixtures upon heating up to 80 °C with the same scanning rate. Raman spectra of crystalline ibuprofen, menthol, and several IBP:M mixtures were plotted in Figure A1 in Appendix A.

Raw spectra collected upon heating were converted into reduced intensity and the quasielastic scattering of each spectrum was integrated in the very low-frequency range between (ω < 30 cm^−1^) for obtaining the temperature dependence of the quasielastic intensity I_QES_(T) which was used to accurately determine melting temperatures. This method was firstly described for the mixture corresponding to X_IBP_ = 0.3 in Figure 2. Figure 2a presents the temperature dependence of the LFRS while the temperature dependence of the quasielastic intensity I_QES_(T) was plotted and compared with the DSC trace in Figure 2b. The single and sharp endotherm was observed at the same temperature (T ≈ 19 °C) as the sudden increase in the quasielastic intensity. It is the indication that this mixture is very close to the eutectic composition, given that the shoulder detected on high-temperature side of the endotherm can be interpreted as arising from a small amount of crystalline matter in excess with respect to the eutectic mixture. At this concentration (X_IBP_ = 0.3), both crystalline components (ibuprofen and menthol) melt strictly at the same temperature (T_E_ ≈ 19 °C). Consequently, the signature of the eutectic melting must be observed in all mixtures, even for compositions far from the eutectic mixture. The same analysis performed on a ibuprofen-rich mixture (X_IBP_ = 0.8) is presented in Figure 3. In this case, the proportion of eutectic mixture is relatively weak compared to that of ibuprofen, inducing a less intense endotherm associated with a very small intensity increase in I_QES_ compared to the DSC and Raman signals detected above T_E_. The dissolution of the excess of crystalline material (with respect to the eutectic proportion) starts just above T_E_, as observed in Figure 3b via the broad endotherm ending around 70 °C, rigorously corresponding to the very spread out increase in the quasielastic intensity.

The analysis of the temperature dependence of the LFRS of mixture richer in menthol (X_IBP_ = 0.1) was presented in Figure A2 in Appendix A. I_QES_(T) curves were systematically plotted for mixtures corresponding to hypo-eutectic and hyper-eutectic concentrations in Figure 4a and Figure 4b, respectively. Two set of experimental points were thus determined and reported in Figure 1, forming the experimental phase diagram of the binary system. It is clearly observed that the eutectic composition (X_IBP_ ~ 0.3) is slightly shifted with respect to that corresponding to ideality, accompanied with a slight depressed eutectic temperature (ΔT ≈ 6°C). Despite the eutectic mixture is liquid at room temperature, the IBP:M system cannot strictly be considered as deep eutectic. The negative deviation from ideality is mainly observed in the hyper eutectic concentration range, in the close neighboring of the eutectic concentration. A detail inspection of Figure 4a shows a two-step increase in I_QES_(T) for the mixture X_IBP_ = 0.1, reflecting the melting of the eutectic mixture followed by the dissolution of the excess of crystalline menthol. This supports the absence of solid solution on the menthol-rich side, as previously suggested [14,18].

#### 3.1.2. Stability Degree of Ibuprofen versus Location in the Phase Diagram

Transdermal delivery of ibuprofen requires avoiding unexpected recrystallization. In this context, after melting crystalline IBP_X_:M_(1−X)_ mixtures, samples were rapidly cooled down to −100 °C for collecting LFRS upon a second heating in order to determine T_g_ and to detect possible recrystallization signatures. A set of LFRS collected upon the second heating of three mixtures (X_IBP_ = 0.2, 0.3, 0.6) was plotted in Figure A3 in Appendix A. The DSC traces were compared with the I_QES_(T) curves for the same mixtures in Figure 5. Signatures of crystallization were clearly observed, both from DSC traces and LFRS. It is worth noting that the eutectic composition (X_IBP_ = 0.3) exhibits very light crystallization signatures, compared with other compositions in Figure 5 and Figure A3. For menthol-rich mixtures (X_IBP_ = 0.2, 0.3) two exothermic peaks were observed corresponding the successive recrystallization of menthol and ibuprofen detected in Figure 5, and better highlighted in Figure A4 in Appendix A for X_IBP_ = 0.2. Figure 5c show that only IBP recrystallizes in IBP-rich mixtures (X_IBP_ = 0.6), and the degree of crystallization increases with the IBP content.

#### 3.1.3. Analysis of H-Bonding in IBP:M Mixtures

H-bonding in IBP:M mixtures can be analyzed both in the C=O and O–H stretching regions. The spectra taken in various IBP:M mixtures were, respectively plotted in these regions in Figure 6a and Figure 6b. From the chemical structures of ibuprofen and menthol plotted in Figure 1, menthol and ibuprofen can be identified as H-bond donor (HBD) and H-bond acceptor (HBA), respectively [14]. Consequently, only the C=O stretching region of IBP contains information about C=O…H H-bonding. Figure 6a highlights the emergence of a Raman band around 1700 cm^−1^ by addition of menthol to the detriment of the band located at 1650 cm^−1^ in IBP. A concomitant intensity increase is observed in the O–H stretching region around 3450 cm^−1^. These spectral features can be interpreted as resulting from the formation of O–H(M)…O=C(IBP) H-bonding interactions between heteromolecules to the detriment of C=O…H homomolecular interactions between IBP molecules. The fitting process of the C=O stretching region is shown in Figure A5a in Appendix A. This procedure provides information on the strength of H-bonding interaction via the frequency of the stretching band and its temperature dependence. Positive temperature dependence in the stretching band frequency is generally considered as typical behavior of the stretching of intramolecular bonds involved in H-bonding [4,36]. The temperature dependences of the C=O stretching band related to C=O…H H-bonding are plotted in Figure 7a for IBP and various amorphous mixtures. A quick look at Figure 7a clearly shows that the band frequency is significantly lower in IBP than in IBP:M mixtures. The increase in the band frequency and the concomitant less-marked positive temperature dependence observed by adding a small amount of menthol confirm the disorganization of the H-bond network of IBP to the detriment of weaker H-bonding interactions between heteromolecules. The two-step temperature behavior observed for hyper eutectic mixtures (X_IBP_ ≥ 0.3) reflects the competition between two types of H-bonding (homomolecular and heteromolecular) interactions. The weaker H-bonding interactions between ibuprofen and menthol break upon heating making it possible dimer formation of ibuprofen molecules.

Heteromolecular H-bonding interactions are clearly revealed by the enhanced intensity in the O–H stretching region, centered around 3400 cm^−1^ only observed in mixtures (see Figure 6b) not existing in pure compounds. This additional intensity gives rise to a broad band characterized by a positive temperature dependence of its frequency (see Figure 7b). Contrasting to the other broad band detected at lower frequencies which is temperature independent in co-amorphous blends (see Figure 7b), the high frequency band can be interpreted as reflecting weak O–H_(M)_ …O_(IBP)_ H-bonding between menthol and ibuprofen molecules [4].

Although the low degree of crystallization detected upon heating IBP-M mixtures close to the eutectic composition, there is no domain in the phase diagram that guarantees the absence of unexpected recrystallization of the amorphous blends at room temperature. These cold recrystallization phenomena are promoted by the low T_g_ values of the various amorphous blends, systematically below −50 °C, as shown in Figure A6 in Appendix A, and the presence of very weak H-bonds between ibuprofen and menthol molecules.

### 3.2. Dissolving IBP in Thymol–Menthol DES

Thymol–menthol was initially known as DES, while more recent investigations have revealed the formation of co-crystals for various mixture compositions [20]. However, the eutectic mixture is in the liquid state at room temperature and both thymol and menthol are recognized to be permeation enhancers. Consequently, Raman investigations were performed on an amorphous blend composed of 25 wt% of IBP dissolved in the eutectic mixture of thymol–menthol, in order to monitor possible unexpected recrystallization of IBP. The low-frequency spectra of IBP dissolved in the DES were collected upon heating at 1 °C/min from −120 °C up to 90 °C (about 14 °C above the melting temperature of IBP) and plotted in Figure 8a after transformation into reduced intensity. No trace of crystallization can be detected in this figure. The absence of crystallization signatures reflects the stability of amorphous IBP dissolved in the eutectic composition of the DES. This stability of amorphous IBP cannot be explained by T_g_ of the ternary mixture significantly higher than T_g_ in IBP:M mixtures, since the plot of I_QES_(T) in Figure 8b shows T_g_ < −50 °C.

Molecular H-bonding associations in the amorphous ternary mixture (thymol–menthol–ibuprofen) are analyzed from the Raman spectra of the C=O and O–H stretching vibrations plotted in Figure 9a,b, respectively. Figure 9a shows that the H-bond network of IBP is disrupted by dissolving IBP in thymol:menthol amorphous blend, via the shift towards the high frequencies of the 1650 cm^−1^ band, as observed in the IBP3:M7 amorphous blend. However, Figure 9b shows that the O–H…O H-bond network in thymol:menthol remains similar after dissolving IBP, maintaining stable amorphous IBP by preventing the formation of the H-bond network of crystalline IBP.

### 3.3. Co-Amorphous Ibuprofen–Amino Acid Systems

Recent studies [17] have shown the high capabilities of arginine (ARG) to form stable co-amorphous systems with indomethacin by milling and spray-drying, compared with other amino acids (AAs). To better understand the properties of AAs that promote the formation of stable co-amorphous blends, binary mixtures based on ibuprofen and arginine were analyzed. Given that T_g_ of ibuprofen is well below room temperature (~−50 °C), by contrast to indomethacin (T_g_ = 42 °C), cryomilling was used to successfully prepared co-amorphous blends. Low-frequency Raman investigations were firstly performed on co-amorphous formulations upon heating from −100 °C to temperatures above melting of IBP (>76 °C), in order to assess the degree of stability and T_g_ of the formulations. In a second step, Raman investigations were also performed at higher frequencies in the C=O stretching region for analyzing H-bonding molecular associations.

#### 3.3.1. Low-Frequency Raman Investigations

The cryomilled IBP_x_ARG_(1-x)_ mixtures were directly loaded in spherical pyrex containers and placed under the N_2_ gas stream of the Oxford device regulated at −100 °C. The spectra collected at −100 °C were converted in reduced intensity and plotted in Figure 10 for mixtures whose IBP content is ranging between 0.9 and 0.5.

Figure 10 shows that mixtures have been amorphized by cryomilling except for high IBP content (X_IBP_ = 0.9). The spectrum of the IBP_0.9_ARG_0.1_ mimics the broadened phonon peaks of crystalline IBP with the presence of the enhanced low-frequency intensity with respect to the crystal spectrum, mostly inherent to the ARG amorphization. For lower IBP contents, the spectra of mixtures reflect the density of vibrational states without detection of any trace of phonon peaks. This indicates that the cryomilled mixtures (X_IBP_ < 0.9) were successfully amorphized. The stability of the co-amorphous blends was analyzed by heating from −100 up to 100 °C, knowing that IBP melts at 76 °C. The quasielastic intensity (I_QES_) was plotted against temperature in Figure 11a for various IBP:ARG co-amorphous blends. I_QES_ was calculated by integrating the low-frequency spectra collected upon heating and plotted in Figure A8 in Appendix B for X_IBP_ = 0.5 and 0.8.

The I_QES_(T) plots of co-amorphous blends are compared with that obtained by heating glassy IBP. Figure 11a reveals the partial recrystallization of IBP-rich co-amorphous blends. For $X_{IBP}\geq 0.5$, the I_QES_(T) plots of mixtures do not exhibit any trace of recrystallization. After a first heating up to 100 °C, IBP:ARG blends were recooled down to −100 °C at 6 °C/min (maximum cooling rate) and spectra were taken again during a second heating ramp at 1 °C/min. The I_QES_(T) plot obtained by integrating the LFRS collected in the second heating ramp is compared in Figure 11b to that corresponding to the first heat for the IBP_0.8_:ARG_0.2_ blend which exhibits the more marked crystalline signatures. It is clearly observed that no trace of crystallization can be detected upon heating the solid co-amorphous blend prepared by cooling the melt, contrasting with the I_QES_(T) plot of the first heating ramp. These two different temperature behaviors of I_QES_ reveal two different amorphous states.

#### 3.3.2. Raman Investigations in the C=O Stretching Region

It was shown in Figure 7a that the C=O stretching band detected around 1650 cm^−1^ in IBP reflected molecular associations via H-bonding. Consequently, this region was investigated for analyzing the H-bonded network of IBP within various IBP_X_:ARG_(1−X)_ blends. Spectra of C=O stretching vibrations were plotted in Figure 12.

The intensity of the 1650 cm^−1^ band reflecting C=O…H H-bonding interactions in IBP is clearly decreasing with the increase in the ARG content. The 1650 cm^−1^ band is transforming into a very broad component as X_IBP_ decreases toward 0.5, which is induced by the shoulder growing on the high frequency side of the band. In the spectrum of IBP_0.4_:ARG_0.6_, there is no remaining trace of additional Raman bands reflecting C=O…H molecular associations between IBP molecules. This behavior can be explained by the disruption of the H-bond network as the ARG content increases. The crystallization of IBP is therefore inhibited by the disruption of its H-bond network inherent to the addition of ARG.

The low-frequency and the C=O stretching spectra taken at −100 °C just after cryo milling and after cooling the liquid were compared in Figure 13 for the blend corresponding to X_IBP_ = 0.8, in order to understand the origin of the stability of the amorphous blend in the second heating ramp. Figure 13a shows similar molecular organization in the short-range in the two amorphous states prepared by direct cryomilling or by cooling the melt, via the rigorous superimposition of the low-frequency spectra. It can be noticed that the presence of a small amount of ARG (X_ARG_ = 0.2) significantly modifies the low-frequency spectrum of IBP and therefore the organization of IBP molecules. By contrast to the low-frequency spectrum, the C=O stretching region distinctive of C=O…H molecular associations between IBP molecules is irreversibly modified after the first heating of the cryomilled blend into the liquid state. The intensity of the 1650 cm^−1^ band was measured and plotted against temperature in Figure 14 for the first heating ramp. Figure 14 reveals a drastic intensity decrease in the band above 30 °C, at the same temperature as the increase in I_QES_(T) observed in Figure 11b, reflecting the melting of crystallized IBP. The low melting temperature of the blend, detected well below the melting temperature of IBP (76 °C), indicates recrystallization in a nanocrystalline state. The 1650 cm^−1^ band being closely related to the H-bond network of IBP, its intensity decrease with the addition of ARG reflects the disruption of the H-bond network of IBP. The disappearance of this band indicates the irreversible and complete disruption of the H-bond network of IBP upon heating above 30 °C, inhibiting the crystallization of IBP after cooling the melt.

## 4. Discussion

The present study explores different routes for improving the bioavailability of IBP used for transdermal administration inherent to the physical state of the API. IBP was selected as a model drug because of its poor solubility in aqueous media leading to poor bioavailability, and therefore requiring amorphization of the drug for achieving better therapeutic action. The glass transition temperature of IBP is well below room temperature (T_g_~−50 °C) promoting topical administration in its undercooled liquid state but making IBP highly metastable with easy recrystallization at room temperature. Additionally, the poor skin permeability of IBP limits its administration from transdermal delivery systems. Recrystallization of IBP within formulations would be an additional obstacle to the skin penetration of the API. In this context, the results presented above can be discussed to better understand the stabilization mechanism of the undercooled liquid state of IBP.

In a first step, menthol–ibuprofen eutectic system was carefully investigated for analyzing the stability of the undercooled liquid state of IBP within the formulation. The physical state of IBP was monitored upon two successive heating ramps at 1 °C/min of various IBP_X_:M_(1−X)_ mixtures. The first heating ramp performed from the crystalline state of the mixture has provided data used for plotting the phase diagram of the binary mixture. Results are very similar to those recently published with the eutectic composition slightly below X_IBP_ = 0.3, and the eutectic temperature very close to 20 °C. The aim of this study focused on the thermal stability of the liquid formulation. Consequently, a special attention was given to the second heating ramp for measuring T_g_, monitoring the physical stability of IBP and probing various (homomolecular, heteromolecular) H-bonding networks. It was found that any mixtures undergo successive partial recrystallization of menthol (around −25 °C) and ibuprofen (around 0 °C for mixtures close to the eutectic composition and around room temperature for ibuprofen-rich mixtures, see Figure 5).

The recrystallization can be obviously explained from consideration of H-bonding interactions. The C=O stretching region can be used for monitoring H-bonding between ibuprofen molecules via the 1650 cm^−1^ band and heteromolecular IBP–M H-bonding interactions via the 1700 cm^−1^ band. This spectral region was plotted at various temperatures for two binary mixtures in Figure 15. For menthol-rich mixtures close to eutectic composition (see Figure 15a), the two types of H-bonding can be detected in the low-temperature range (−50 °C) from the presence of the two bands. The weak intensity of the 1650 cm^−1^ band is related to the low content of ibuprofen in the mixture. However, the detection of this band indicates that all IBP molecules are not interacting with menthol molecules, and some dimers of IBP molecules are preserved in the liquid formulations at low temperatures. The higher intensity of the 1700 cm^−1^ band is induced by the contribution of the Raman band of pure menthol and the additional contribution inherent to ibuprofen–menthol H-bond interactions (see Figure 6a). Only a few menthol molecules are associated with ibuprofen molecules via H-bonding, making recrystallization of not associated menthol molecules possible. The small amount of IBP dimers detected at 0 °C around 1650 cm^−1^ makes partial recrystallization of IBP possible. Upon further heating, the disruption of the two types of H-bonding networks is observed via the disappearance of the 1650 cm^−1^ band and the shift towards the high frequencies of the 1700 cm^−1^ band. For ibuprofen-rich mixtures, e.g., X_IBP_ = 0.7 (Figure 15b), most of menthol molecules are H-bonded with ibuprofen molecules. As a consequence, menthol does not recrystallize as observed in Figure 5c for X_IBP_ = 0.6 via the detection of only one exotherm with a concomitant decrease in the quasielastic intensity slightly above 25 °C, corresponding to the partial recrystallization of IBP (see Figure A7 in Appendix A). At low temperature (−10 °C), Figure 15b shows a very weak amount of ibuprofen dimers, increasing with temperature to the detriment of heteromolecular H-bonding. The formation of IBP dimers promotes ibuprofen recrystallization, shifted towards higher temperatures than that determined in menthol-rich mixtures (below 0 °C).

An alternative to the formation of a eutectic IBP:M binary system for avoiding recrystallization of IBP within the liquid formulation is to dissolve IBP in natural deep eutectic solvents (DES). Knowing the remarkable properties of terpenes for enhancing permeation of drugs across the skin [14,19], and the opportunity to form DES by mixing them [10,19], the dissolution of IBP in the eutectic composition of thymol:menthol system was carefully analyzed. Despite the recent evidence of cocrystal formation in its phase diagram [20], the mixture prepared in equimolar proportion is liquid at temperatures well below room temperature. The maximum mass of IBP which can be dissolved in thymol:menthol (1:1) was estimated to 25 wt% with respect to the DES. Figure 8 shows the total absence of crystallization signature of IBP dissolved in the DES, via the absence of phonon trace in the LFRS and the monotonic temperature dependence of I_QES_(T), despite the low T_g_ (~−70 °C) of the formulation induced by the plasticization effect of the DES. The stability of IBP can be explained by the absence of the 1650 cm^−1^ band reflecting the absence of ibuprofen dimers dissolved in the DES inherent to the H-bond formation between IBP and the DES components. The C=O stretching region, plotted at −50 and 50 °C in Figure 16 shows the absence of the 1650 cm^−1^ band distinctive of IBP dimers. By contrast, the 1700 cm^−1^ band shifting towards the high frequencies upon heating, shows H-bonding interactions between ibuprofen and menthol/thymol molecules. This study shows that dissolving IBP in the DES composed of terpenes should be a better route for avoiding IBP recrystallization than forming eutectic mixtures (ibuprofen–menthol or ibuprofen–thymol).

In a second step, the stability of amorphous IBP was analyzed within co-amorphous blends prepared by cryomilling ibuprofen with amino acids. Arginine was selected because of its remarkable properties for stabilizing amorphous indomethacin (IMC). However, T_g_ of IBP is significantly lower than T_g_ of IMC (T_g_(IMC)–T_g_(IBP) ≈ 90 °C). T_g_ values of various mixtures determined from low-frequency Raman data are compared, in Figure A9 in Appendix B, to the T_g_ curve calculated from the Gordon-Taylor law. This figure shows that T_g_ of IBP-rich mixtures (X_IBP_ ≥ 0.4) is below room temperature. However, no recrystallization was detected in the mixture prepared by cryomilling ARG and IBP in equimolar proportion. Additionally, no recrystallization was detected in previously melted IBP-rich mixtures. Consequently, this study reveals that T_g_ is not the relevant parameter for predicting the physical stability of amorphous IBP within co-amorphous blends by contrast to the results recently obtained from investigations in AA:IMC blends [13,16,37]. For IBP-based mixtures, the inability to form ibuprofen dimers by adding ARG seems the main cause for inhibiting IBP recrystallization, even for low content of ARG, after melting of the cryomilled mixture. After melting, the short-range molecular organization is similar to that obtained just after cryomilling, but no trace of IBP dimers can be detected after melting contrary to what is observed after milling. Consequently, the inhibition of IBP dimer formation by heteromolecular H-bonding formation seems to be the required condition for avoiding partial recrystallization of ibuprofen.

## 5. Conclusions

The main goal of this study was to analyze the physical stability of amorphous IBP within various formulations designed for transdermal administration. Most of investigations reported in the literature aims to estimate the efficacy of IBP for skin penetration [23,24,25] without consideration of the physical state of IBP. However, unexpected recrystallization of IBP should have negative consequence on skin penetration of the API. This paper reports the first analyses of the physical stability of ibuprofen in topical formulations. It was shown that partial recrystallization of ibuprofen was detected in the two types of analyzed formulations. However, the method used for preparing the two types of preparation has a direct impact on the stability of amorphous ibuprofen. It can be concluded that dissolving IBP in terpenes-based DES and preparing co-amorphous IBP:ARG blends obtained by melting are the best routes for avoiding the partial recrystallization of IBP. Despite the antiplasticizing effect of ARG leading to T_g_ of IBP:ARG mixtures well above T_g_ (~−60 °C) of the ibuprofen–menthol:–hymol formulation, the stabilization mechanism of IBP is more likely related to the inhibition of IBP dimer formation by heteromolecular H-bond associations between IBP and ARG or DES components. Consequently, the presence of IBP dimers within formulations can be considered as a precursor of unavoidable recrystallization. This study shows that ibuprofen easily recrystallizes within these two analyzed formulations. This suggests the possible recrystallization of ibuprofen within various types of topical formulations, and careful attention should be paid to this issue.

## Figures and Tables

**Figure 1 pharmaceutics-15-01710-f001:**
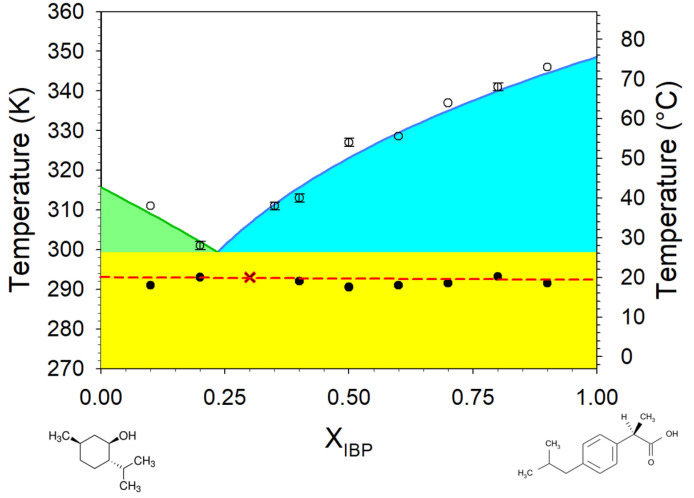
Theoretical (line) and experimental (full and open circles) phase diagram of IBP:ARG. Chemical structures of IBP and M are plotted at the bottom of the diagram. The red cross indicates the composition and the melting temperature of the eutectic mixture from experimental determination. The red dotted line indicates the eutectic temperature.

**Figure 2 pharmaceutics-15-01710-f002:**
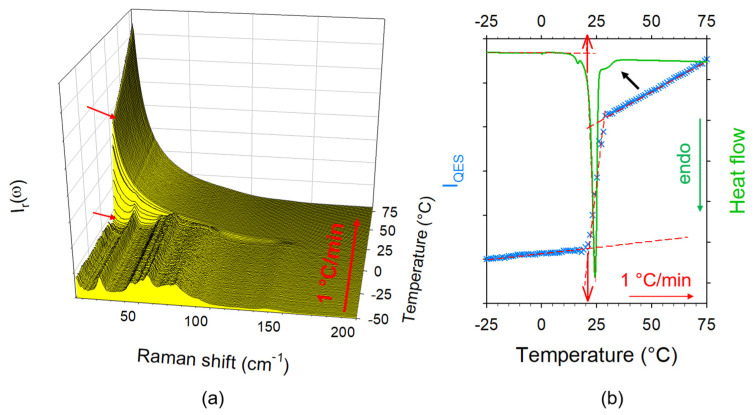
Heating process of crystalline IBP_X_:M_(1−X)_ (X_IBP_ = 0.3); (**a**) temperature dependence of the low-frequency—the red arrows localize the beginning and the end of the melting; (**b**) comparison of I_QES_(T) plot with the DSC trace. The red arrows show the comparison between Tm determined from DSC and LFRS data. The black arrow highlights the dissolution of the very small amount of crystalline matter in excess with respect to the eutectic composition.

**Figure 3 pharmaceutics-15-01710-f003:**
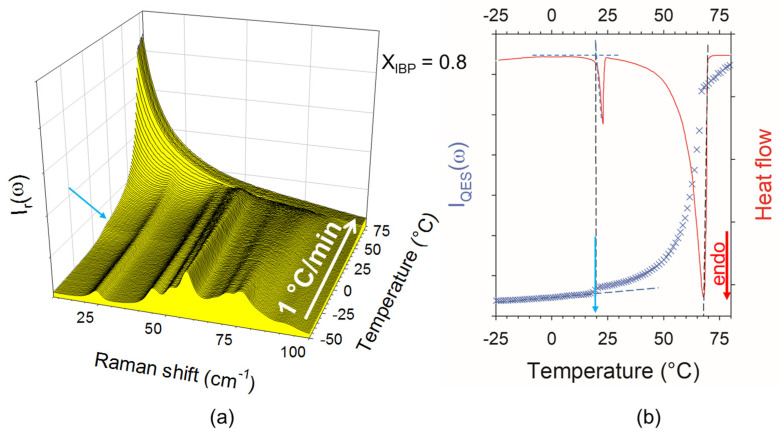
Heating process of crystalline IBP_X_:M_(1−X)_ (X_IBP_ = 0.3); (**a**) temperature dependence of the low-frequency; (**b**) comparison of I_QES_(T) plot with the DSC trace. The blue arrow localizes the beginning of the I_QES_ increase corresponding to the endotherm of the eutectic melting.

**Figure 4 pharmaceutics-15-01710-f004:**
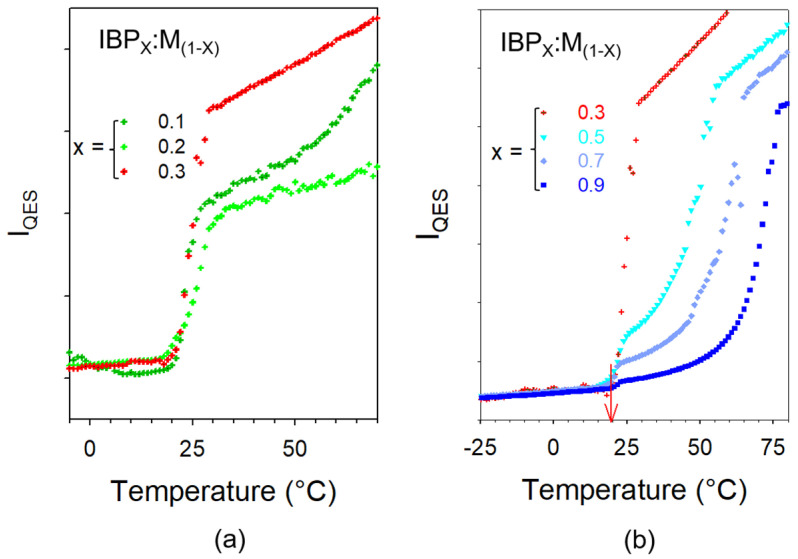
I_QES_(T) plots for the various IBP_X_:M_(1−X)_ mixtures; (**a**) in hypo-eutectic mixtures; (**b**) in hyper eutectic mixtures; the red arrow indicates the melting temperature of the eutectic composition.

**Figure 5 pharmaceutics-15-01710-f005:**
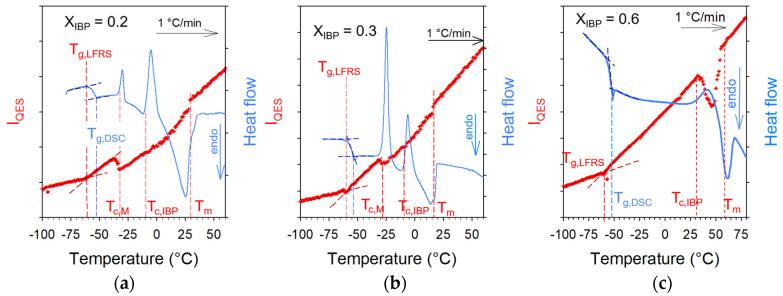
Analysis of phase transformations of various IBP_X_:M_(1−X)_ mixtures upon heating from the glassy state (**a**) X_IBP_ = 0.2, (**b**) X_IBP_ = 0.3, (**c**) X_IBP_ = 0.6; vertical dashed lines localize temperature of glass transition determined from LFRS and DSC data, temperatures of crystallization of M and IBP and melting temperature.

**Figure 6 pharmaceutics-15-01710-f006:**
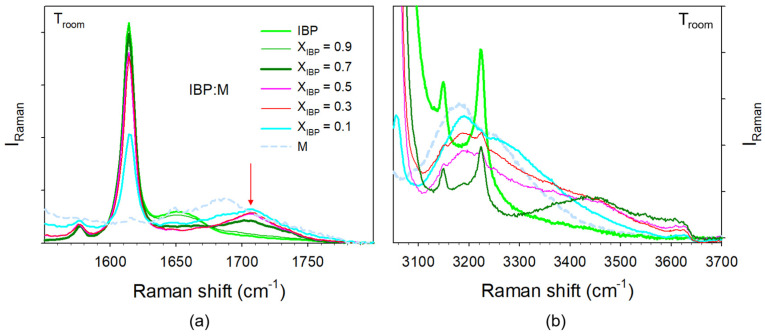
Raman spectra taken at room temperature in various IBPX:M(1−X) mixtures (**a**) in the C=O stretching region, the red arrow localizing an additional Raman band only existing in the mixtures; (**b**) in the O–H stretching region.

**Figure 7 pharmaceutics-15-01710-f007:**
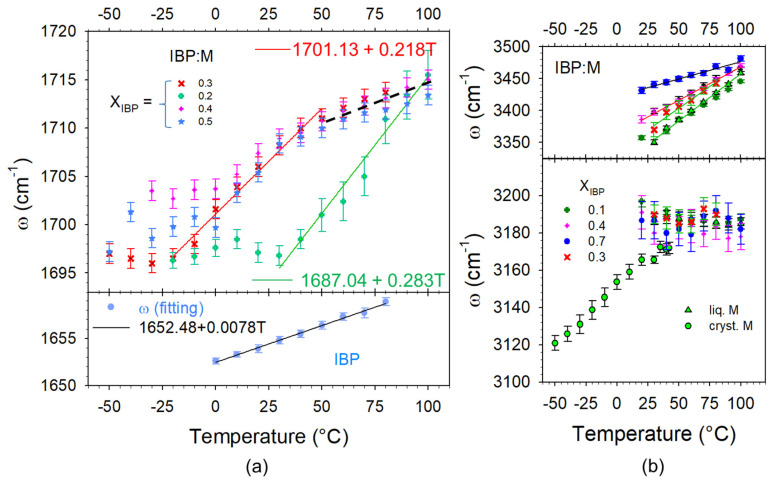
Temperature dependences in various amorphous mixtures of the frequencies of Raman bands involved in H-bonding, (**a**) C=O…H and (**b**) O–H …O molecular associations; lines correspond to fitting procedures of ω(T) by linear regressions, black dashed line is guide for eyes.

**Figure 8 pharmaceutics-15-01710-f008:**
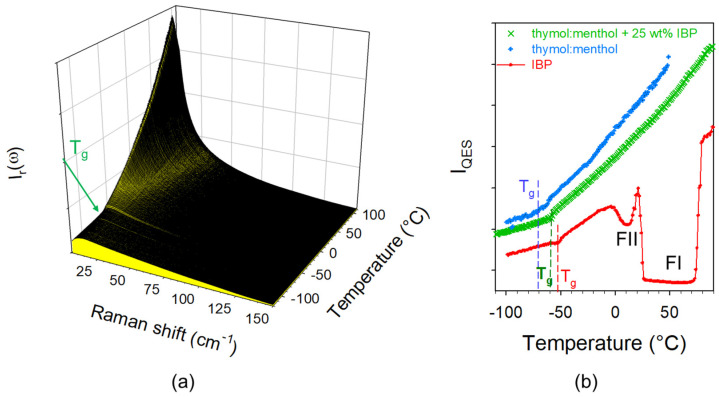
Thermal stability of amorphous IBP dissolved in thymol:menthol analyzed from (**a**) the LFRS of the ternary mixture collected upon heating at 1 °C/min, (**b**) I_QES_(T) plot of the ternary mixture determined from integrating LFRS, compared with I_QES_(T) plots of thymol:menthol and glassy IBP obtained by quenching the liquid state. FII and FI indicate the metastable form and stable form of crystalline IBP corresponding to minima of _IQES_(T) plot.

**Figure 9 pharmaceutics-15-01710-f009:**
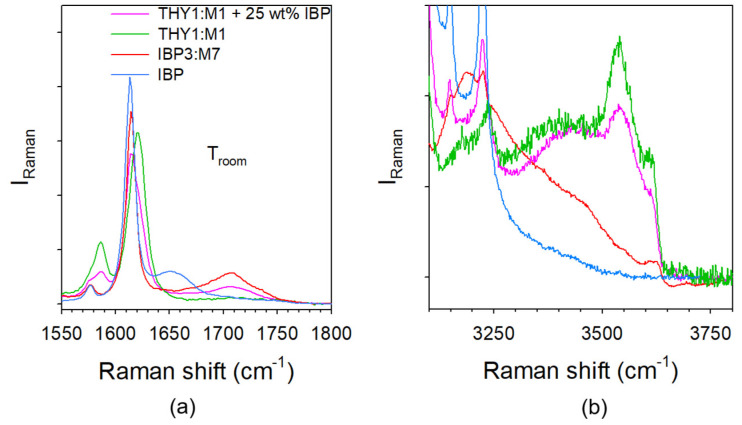
Analysis of H-bonding molecular associations in the ternary thymol:menthol:IBP mixture compared with those in IBP, in thymol:menthol (THY_1_:M_1_) and IBP_3_:M_7_ mixtures (**a**) in the C=O stretching region, (**b**) in the O–H stretching region.

**Figure 10 pharmaceutics-15-01710-f010:**
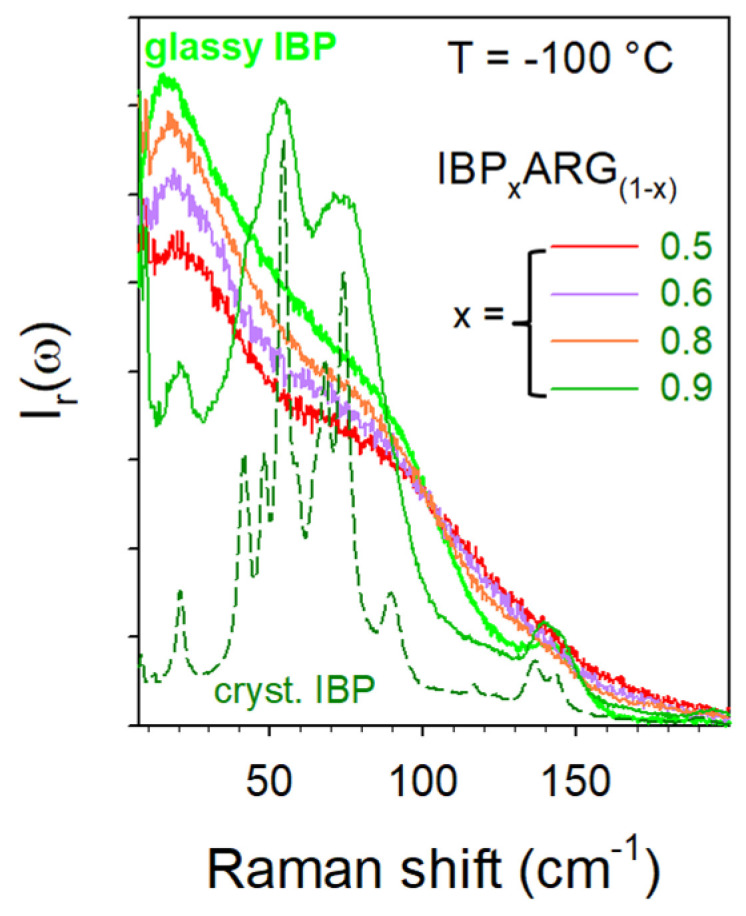
LFRS of IBP_X_:ARG_(1−X)_ mixtures collected at −100 °C just after cryomilling.

**Figure 11 pharmaceutics-15-01710-f011:**
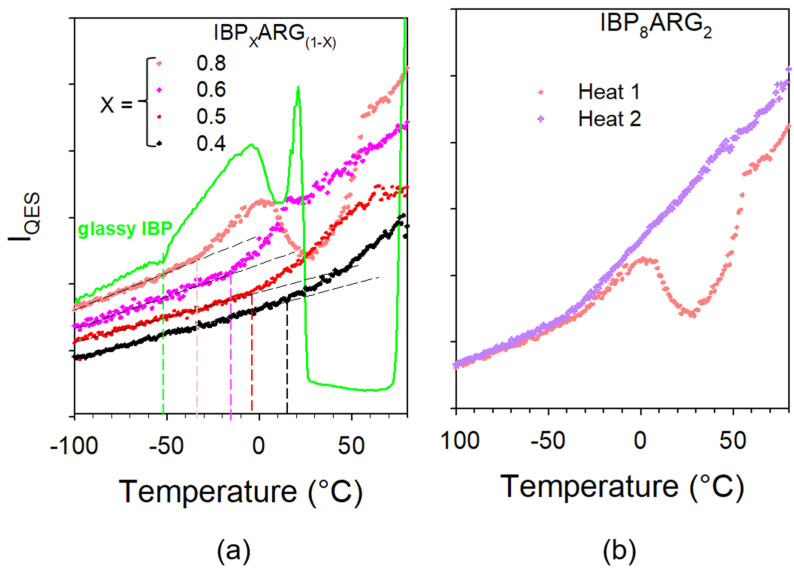
I_QES_(T) plots obtained from (**a**) the 1st heating ramp of various cryomilled mixtures, (**b**) the 1st and 2nd heating ramps of the cryomilled IBP_8_:ARG_2_ mixture. Vertical dashed lines indicate T_g_ values, increasing with X_IBP_ decrease.

**Figure 12 pharmaceutics-15-01710-f012:**
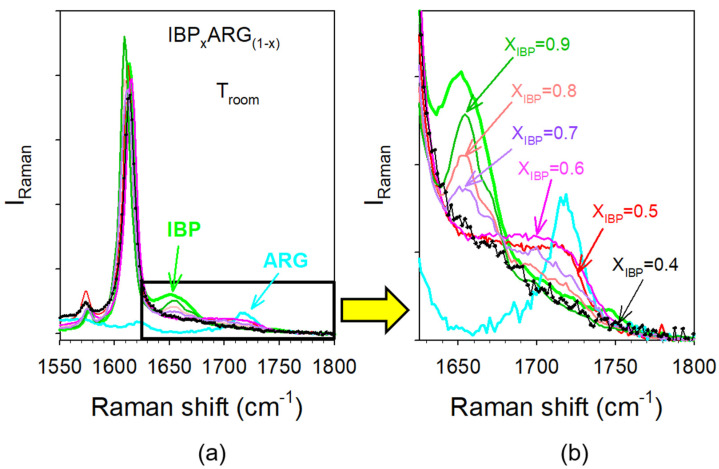
Analysis of H-bonding in various IBP_X_:ARG_(1−X)_ mixtures (**a**) in the C=O stretching region (**b**) in the narrower region centered on Raman bands involved in H-bonding.

**Figure 13 pharmaceutics-15-01710-f013:**
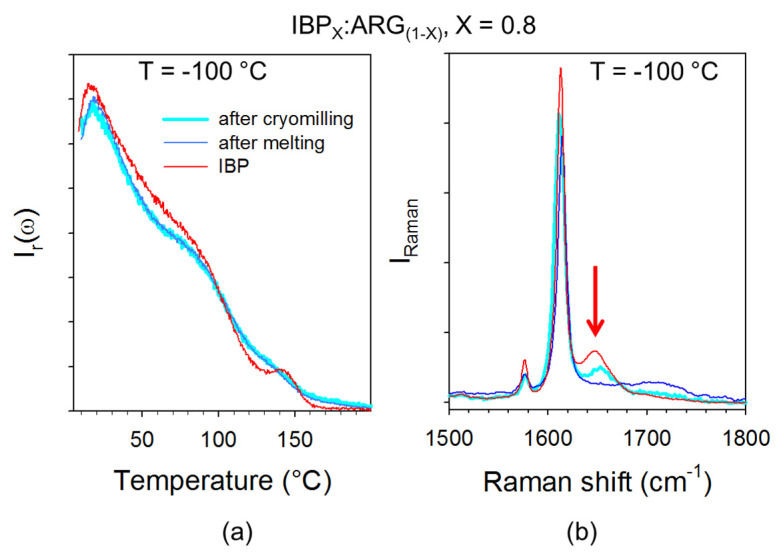
Comparison between Raman spectra of the X_IBP_ = 0.8 blend taken at −100 °C just after cryomilling and after a first heating ramp up to 100 °C (**a**) in the low-frequency region, (**b**) in the C=O stretching region; the arrow localizes the Raman band distinctive of the H-bond network of IBP. These spectra are compared with those of glassy IBP at the same temperature.

**Figure 14 pharmaceutics-15-01710-f014:**
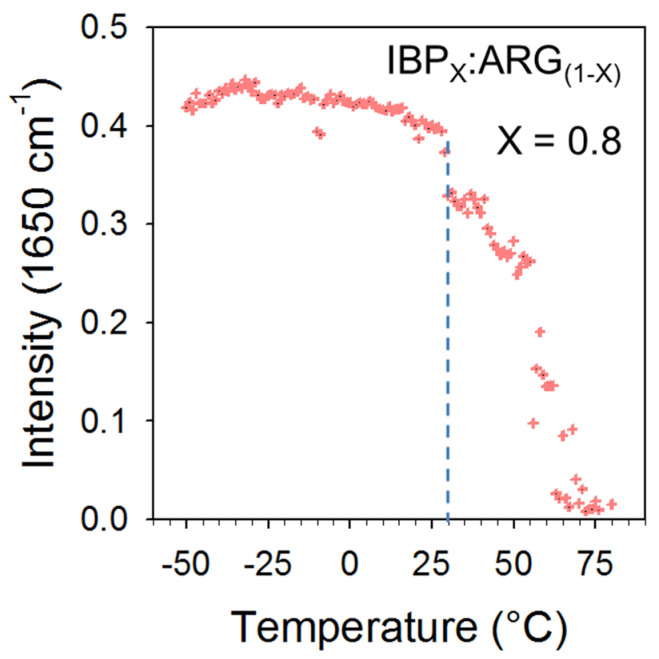
Integrated intensity of the 1650 cm^−1^ band against temperature determined from normalized spectra collected upon a first heating of the X_IBP_=0.8 blend; the vertical dashed line roughly localizes the temperature at which the 1650 cm^−1^ band begins to disappear.

**Figure 15 pharmaceutics-15-01710-f015:**
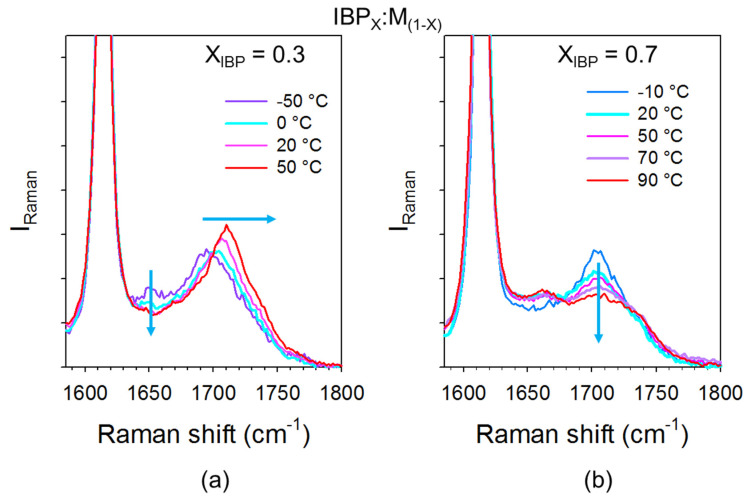
Spectra of C=O stretching vibrations of IBP_X_:M_(1−X)_ mixtures collected at various temperature upon heating from the glassy (**a**) for menthol-rich mixture, (**b**) for ibuprofen-rich mixture. The arrows highlight the main changes in the spectra.

**Figure 16 pharmaceutics-15-01710-f016:**
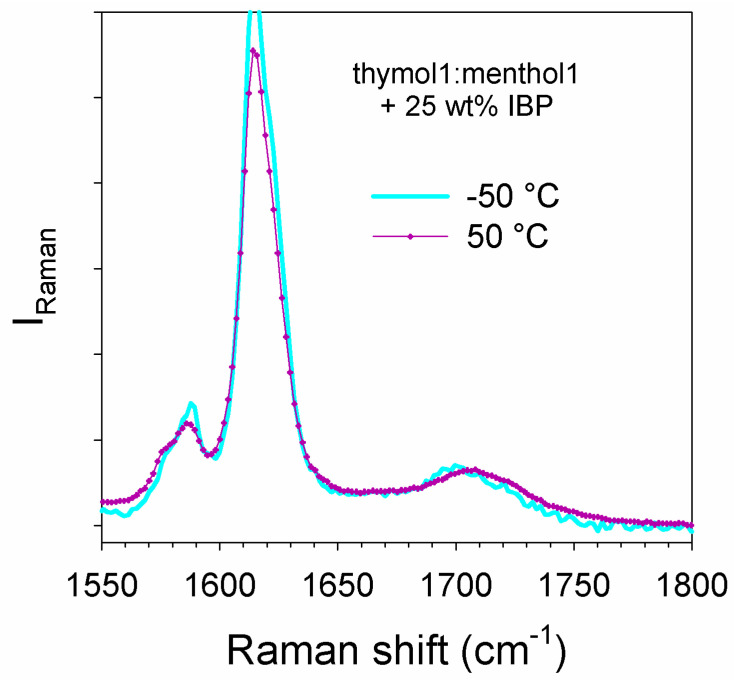
Temperature dependence of the C=O stretching spectrum of the ternary mixture prepared by dissolving 25 wt% of IBP in the binary mixture thymol–menthol in equimolor proportions.

## Data Availability

Data are shown within the article.

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
