# Peer review of "Co-Amorphous Versus Deep Eutectic Solvents Formulations for Transdermal Administration"

_pharmaceutics, 2023, doi:10.3390/pharmaceutics15061710_

Round 1
Reviewer 1 Report
The authors present intriguing methods for improving ibuprofen's thermal stability in topical formulations using terpenes-based DES and arginine-based co-amorphous blends. Raman spectroscopy findings suggest that heteromolecular hydrogen-bonding formation is critical to preventing ibuprofen recrystallization. The paper demonstrates considerable potential. However, to fully realize this potential and strengthen the manuscript, it requires minor revisions as outlined in the provided comment:
1. Lines 35-38: The explanation on how solid dispersions were developed to increase Tg and the challenge of ensuring the stability of amorphous drugs within the polymer carrier is well explained. However, it would be helpful to elucidate more on the 'specific experimental methods' mentioned which are apparently not easy to implement.
2. Lines 39-44: The authors introduce the idea of using amino acids as co-amorphous stabilizers, but it would be beneficial if they could elaborate more on why amino acids are effective in this role.
3. Lines 81-85: The statement of the study's aim is clear, but the authors should justify their choice of ibuprofen as the model drug and clarify what they mean by 'the most reliable opportunities'.
4. Line 269-271: The spectral evidence of H-bonding interactions is presented, but it's unclear how the authors are differentiating between intermolecular and intramolecular H-bonds. A more detailed explanation would be beneficial for understanding.
5. Lines 319: The authors claim that no trace of crystallization can be detected, but the methods used to confirm this could be described more clearly. Are there specific peaks or signatures they are looking for? What other techniques could have been used to confirm this result?
6. In conclusion, the authors state the main goal of the study and the primary conclusions, but they don't elaborate on the implications of the study's findings. They should consider addressing the importance of this research in the field of drug delivery systems and how it could possibly be utilized in future research or development.
Overall, the English language used in this conclusion is of high quality. The sentences are well-structured, and the vocabulary is appropriate for the subject matter.
Reviewer 2 Report
The manuscript of Guinet et al. contains noticeable information on the comparison of transdermal formulations. Though the current version is well-written, some minor modifications seem necessary.
- The sentence in lines 27-29 needs a reformulation. API synthesis and production are fundamental steps in the pharmaceutical industry and have specialties, including, e.g., chemical stability, purity criteria, or impurity profile. Drug formulation is a different science with its own goals, and the reviewer does not consider it appropriate to compare various disciplines in the authors' way.
- In Figure 3a, the red text on the grey-yellow background is hardly readable.
- References do not contain DOIs.
The referee recommends reformulation of some sentences to reduce their lengths.
Reviewer 3 Report
The article "Co-amorphous VS DES formulations for transdermal administration" presents a study on a mixture of ibuprofen (as model drug) and L-arginine or L-menthol (as stabilizing excipient) . It is a valuable study that can be published after authors address the following problems:
Abstract should be checked and revised carefully by briefly introducing the work plan and key findings. Abstracts should highlight the innovation of the article, as often abstract section is presented separately in search engines, it must be able to stand alone as an informative piece. In the abstract, need to focus more on the quantitative information, not qualitative one.
In introduction a stronger recent literature survey is necessary, especially on previous literature reports on the topical drug delivery: doi: 10.1111/php.12117; doi: 10.1080/1539445X.2011.582914; doi: 10.3390/gels9050391; doi: 10.3390/ijms23147752; doi: 10.3390/ijms23084158; doi: 10.3390/ma14226808; doi: 10.3390/ma14216678
Which exact problem was supposed to be solved by the present research? Specify the novelty of this work. Which new achievement(s) was supposed to be obtained by the present research compared to the previous reports.
The conclusion should reflect the heuristic of the study. How is this system a better one? Conclusion section must be reworked to underline the novelty and advantages of this research, with actual numbers.
Minor points:
The English language needs some minor polishing for style and typos.
Do not use abbreviations in the title (vs and DES). The caps VS is versus or has other meaning?
Please use same font for axes among different figures (discrepancies are too high e.g figure 14 vs figure 13).
Reviewer 4 Report
This is an interesting manuscript with several physicochemical techniques for evaluating and comparing co-amourphus vs. eutectic solvents of ibuprofen for transdermal administration.
My comments are:
1. What is the role of each participant that is used in this study?
2. Which are the preformulation studies?
3. In Section 3.1.3, the references are missing.
4. Why do the authors select the transdermal route of administration of Ibuprofen? The added value and comparison with other studies should be addressed in the conclusions.
Round 2
Reviewer 3 Report
Authors significantly improved the manuscripts; all comments were addressed properly.